# The essential role of PRAK in tumor metastasis and its therapeutic potential

Yuqing Wang[1,2,6], Wei Wang[1,6], Haoming Wu[1], Yu Zhou[1], Xiaodan Qin[1], Yan Wang[1], Jia Wu[1], Xiu-Yuan Sun[1], Yan Yang[3], Hui Xu[3], Xiaoping Qian[1], Xuewen Pang[1], Yan Li[1], Zhiqian Zhang ⓘ [4], Jiahuai Han ⓘ [5] & Yu Zhang ⓘ [1,3✉]

Metastasis is the leading cause of cancer-related death. Despite the recent advancements in cancer treatment, there is currently no approved therapy for metastasis. The present study reveals a potent and selective activity of PRAK in the regulation of tumor metastasis. While showing no apparent effect on the growth of primary breast cancers or subcutaneously inoculated tumor lines, *Prak* deficiency abrogates lung metastases in PyMT mice or mice receiving intravenous injection of tumor cells. Consistently, PRAK expression is closely associated with metastatic risk in human cancers. Further analysis indicates that loss of function of PRAK leads to a pronounced inhibition of HIF-1α protein synthesis, possibly due to reduced mTORC1 activities. Notably, pharmacological inactivation of PRAK with a clinically relevant inhibitor recapitulates the anti-metastatic effect of *Prak* depletion, highlighting the therapeutic potential of targeting PRAK in the control of metastasis.

[1] Department of Immunology, School of Basic Medical Sciences, Peking University. NHC Key Laboratory of Medical Immunology (Peking University), Beijing, China. [2] Center of Basic Medical Research, Institute of Medical Innovation and Research, Peking University Third Hospital, Beijing, China. [3] Institute of Biological Sciences, Jinzhou Medical University, Jinzhou, Liaoning, China. [4] Key laboratory of Carcinogenesis and Translational Research (Ministry of Education), Department of Cell Biology, Peking University Cancer Hospital and Institute, Beijing, China. [5] State Key Laboratory of Cellular Stress Biology, Innovation Center for Cell Signaling Network, School of Life Sciences, Xiamen University, Xiamen Fujian, China. [6]These authors contributed equally: Yuqing Wang, Wei Wang. ✉email: zhangyu007@hsc.pku.edu.cn

Tumor metastasis is a major challenge in the clinical management of cancer. Approximately 90% of cancer-related deaths is due to metastatic diseases in vital organs[1]. Metastasis consists of a series of sequential and interrelated steps[2,3]. Tumor cells first detach from the tumor mass, migrate across the adjacent tissue, and invade the blood or lymphatic vessels. In a systemical search for regulators of this initial step, Stoletov and colleagues have reported the identification of a panel of novel genes, whose function is required for productive cancer cell motility in vivo and whose expression is closely associated with metastatic risk in human cancers[4]. Once in circulation, metastatic cells are subject to enormous stresses introduced by shear force, immune attack, and anoikis. The few cells that survive these stresses arrest in the capillary bed, and migrate across the endothelium into the parenchyma. Following extravasation, a cell or cluster of cells can grow into a micrometastasis, die, or become dormant. The colonization and formation of a macroscopic metastatic lesion is ultimately determined by their successful adaptation to the new environment and angiogenesis. Host factors could be particularly important for this step. By screening 810 mutant mouse lines, van der Weyden et al. identified 23 genes critically involved in host control of metastasis[5].

Despite these recent advancements in our understanding of the molecular basis of metastasis, the hypoxia-inducible transcription factor (HIF) remains one of best-studied regulators[6,7]. In a wide range of tumor types, high HIF expression is associated with increased metastasis[6,7]. This heterodimeric transcription factor is composed of an $O_2$-labile HIF-1α subunit and a constitutive HIF-1β subunit. In well-oxygenated cells, $O_2$-dependent hydroxylation of HIF-1α allows its recognition by the von Hippel-Lindau protein (VHL). HIF-1α is thus targeted for ubiquitination and proteasomal degradation. Under hypoxic conditions, HIF-1α hydroxylation is inhibited, resulting in its stabilization and nuclear translocation[6,8]. HIF stabilization may also be achieved through deubiquitination catalyzed by the ubiquitin C-terminal hydrolase-L1 (UCHL1), whose activity is shown to be associated with heightened metastatic potential[9]. In addition to $O_2$-triggered degradation, intracellular levels of HIF-1α are subjected to transcriptional and translational regulation[10].

Increased HIF activity drives tumor progression by regulating the expression of hundreds of genes critically involved in such important aspects of tumor pathobiology as angiogenesis, metabolic reprogramming, cancer stem cell maintenance, immune evasion, metastasis, and resistance to chemotherapy and radiation therapy. With regard to metastasis, HIF signaling influences virtual every step of the cascade[6,7,10]. This master regulator (i) promotes the epithelial-mesenchymal transition (EMT) of tumor cells and the extracellular matrix remodeling to facilitate local tissue invasion; (ii) enhances the passage of detached tumor cells into the lumen of lymphatic or blood vessels by inducing the expression of vascular endothelial growth factor (VEGF)-A, cytokine and chemokine; (iii) confers resistance to anoikis through suppression of α5 integrin signaling[11]; (iv) contributes to tumor cell extravasation at metastatic sites via expression of angiopoietin-like 4 (ANGPTL4) and L1-cell adhesion molecule[12]; (v) facilitates pro-metastatic niche formation through the expression and secretion of lysyl oxidase (LOX) and LOX-like proteins (LOXL2 and 4) and recruitment of bone marrow-derived cells[13,14]; (vi) promotes the formation of macroscopic metastatic lesions by stimulating angiogenesis through VEGF-A expression[15].

The p38-regulated/activated protein kinase (PRAK) belongs to the MAPK-activated protein kinase family. Upon phosphorylation and activation by p38MAPK, atypical MAPK ERK3 and ERK4, or PKA[16], PRAK contributes to the regulation of a range of cellular responses, including Ras-induced senescence[17], nutrient starvation responses[18], proliferation[19], V(D)J recombination[20] and motility[21],

through phosphorylating such substrates as p53, Rheb, FOXO3a, FOXO1, or HSP27. Of particular interest are the paradoxical roles in tumorigenesis[22]. PRAK deficiency is associated with impaired RAS-induced senescence and enhanced skin carcinogenesis following treatment with dimethylbenzathracene[17], suggesting a tumor-suppressive function. In support of this notion, PRAK deletion also leads to accelerated development of hematopoietic malignancy in a mouse model harboring an oncogenic RAS allele, possibly due to hyperactivation of the JNK Pathway[23]. Still another mechanism for PRAK-mediated tumor suppression relies on its capacity to phosphorylate FOXO3a, which enhances miR-34b and miR-34c expression and results in reduced c-MYC protein levels[19]. However, once the tumor is established, PRAK switches to support tumor growth and progression by stimulating endothelial cell migration and angiogenesis[21].

In view of the stress imposed on metastatic cells and the importance of p38-PRAK signaling in the stress response, the present study investigated the potential role of PRAK in another important aspect of tumorigenesis—tumor metastasis. While having no apparent impact on primary tumors, *Prak* deletion abolished the lung metastasis in spontaneous as well as transplanted tumor models. Similar results were obtained with the PRAK inhibitor GLPG0259. The protective effect of PRAK on metastatic tumor cells was primarily achieved through the promotion of HIF-1α translation.

## Results

**Ablation of *Prak* suppresses dissemination but not growth of primary tumors**. To interrogate the function of PRAK in tumor metastasis, we crossbred the *Prak*-deficient mouse with the mouse mammary tumor virus-Polyoma virus middle T-antigen (MMTV-PyMT) mouse, which spontaneously develops breast cancer with the occurrence of lung metastases at the age of 12–14 weeks[24]. No significant difference was observed in the incidence and growth of primary tumors in the breast with or without *Prak* (Fig. 1a). On the other hand, the development of lung metastatic lesions was almost completely inhibited in *Prak*-deficient mice (Fig. 1b). We next generated several *Prak* null mutants from the B16 melanoma cell line using the CRISPR/Cas9 technology (Fig. 1c). Upon subcutaneous inoculation, the *Prak* mutants showed a growth rate comparable to the parent cells (Fig. 1d). In contrast, the mutants yielded a markedly reduced number of tumor nodules in the lung when intravenously administered (Fig. 1e). Accordingly, an increased survival rate was seen for the group inoculated with the *Prak*-null B16 cells (Fig. 1f). The observation was extended to the human melanoma cell A375 and breast carcinoma cell MDA-MB-231, in which *PRAK* expression was knocked down with lentivirus-delivered shRNA (Supplementary Fig. 1a and c). In comparison to the vector-only controls, these shRNA transfectants displayed a much-suppressed capacity to colonize in the lung tissue of SCID mice (Supplementary Fig. 1b and d). Collectively, these results point to a crucial and selective role of PRAK in tumor metastasis.

**PRAK represents a potential target for the intervention of tumor metastasis**. The highly selective action of PRAK in tumor metastasis prompted us to explore its potential in targeted cancer therapy. Over years, several classes of PRAK inhibitors have been developed[25–27]. GLPG0259 is one such compound with an $IC_{50}$ in the nanomolar range[25,28,29]. An early phase II clinical trial demonstrated that it was safe and well tolerated with daily doses over 12 weeks, although the trial was terminated due to the lack of efficacy for the treatment of rheumatoid arthritis[28,29]. To test its impact on tumorigenesis, GLPG0259 was administered into PyMT mice via intraperitoneal injection at a dose of 1 mg/kg

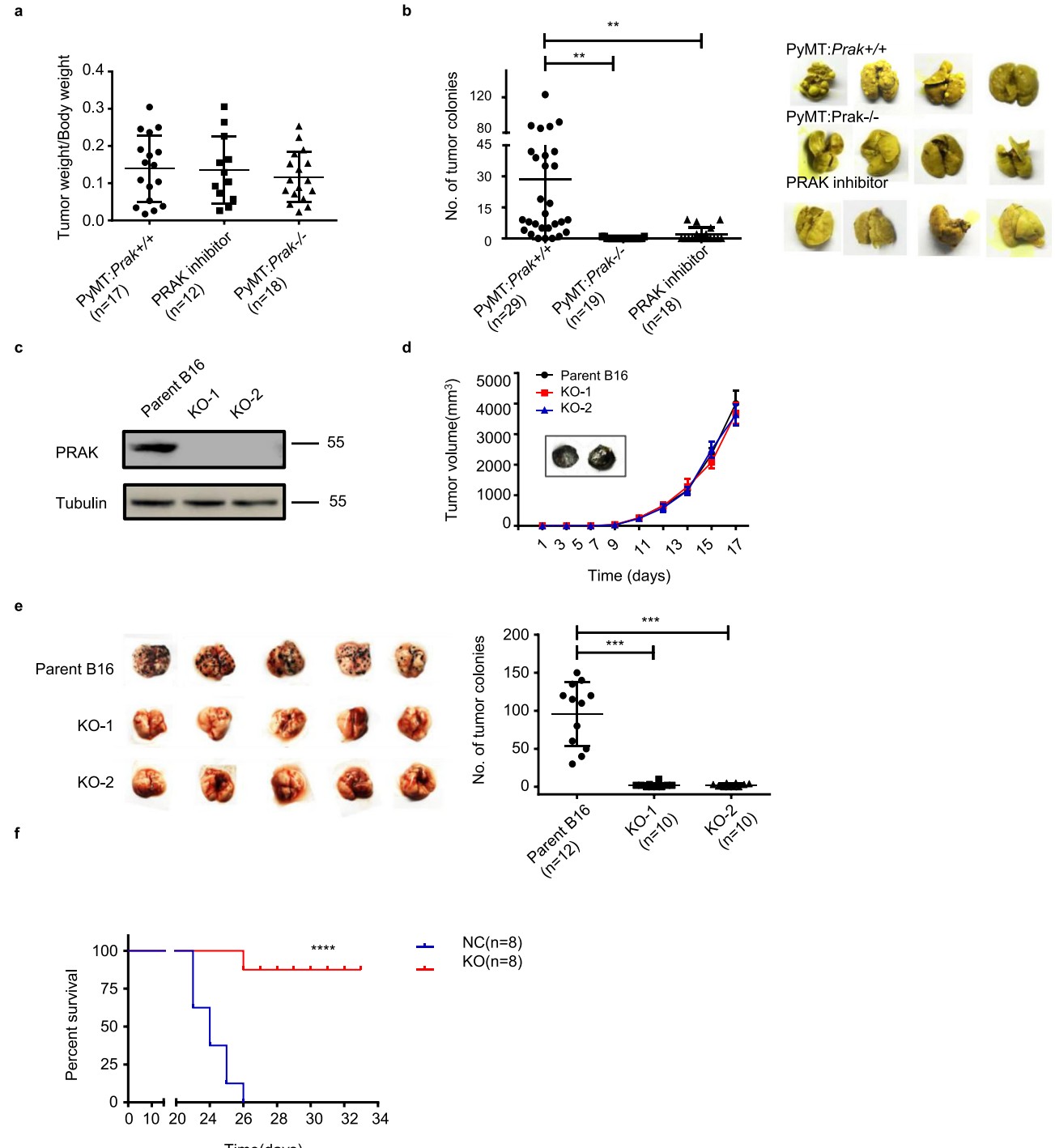

**Fig. 1 *Prak* deficiency impairs distant metastases but not the growth of primary tumors. a**, **b** *Prak* knockout mice were crossbred with MMTV-PyMT mice. The development of mammary tumors and lung metastases were compared among the wild type, *Prak*[−/−], and PRAK inhibitor-treated PyMT mice at week 16. Data are presented as the ratio of mammary tumor vs. body weight (**a**) or the number of metastatic colonies in the lung (**b**). Each symbol represents an individual mouse. **c**–**f** *Prak* knockout (KO) clones were generated from the parent B16 melanoma line using CRISPR/Cas9. Loss of PRAK protein expression was confirmed by Immunoblotting (**c**). The volume of tumors formed by subcutaneously inoculated B16 clones was recorded at different time points. $n = 7$ biologically independent animals were examined. Data are presented as mean ± s.d. with seven mice for each group (**d**). Tumor colonization in the lung was examined at day 16 after intravenous injection. Representative images (**e**, left) and quantification of tumor colonies (**e**, right) are shown. Survival rate was also monitored after intravenous injection of the parent or *Prak* knockout B16 clones (**f**). The histogram bars displayed as mean ± s.d. **\*\***$p < 0.01$, **\*\*\***$p < 0.001$, **\*\*\*\***$p < 0.0001$. *p*-value was determined by two-tailed, unpaired *t*-test (**a**, **b**, **e**) or Log-rank test (**f**).

every two days between 12 and 14 weeks after birth. Consistent with the genetic data described above, pharmacological inactivation of PRAK, while imposing no impact on primary tumors (Fig. 1a), caused a pronounced inhibition of lung metastases

(Fig. 1b). Similar inhibition was observed for GLPG0259 in the formation of lung tumors by intravenously injected B16 (Fig. 2a), A375 (Supplementary Fig. 1b), and MDA-MB-231 (Supplementary Fig. 1d) cells. It is worth mentioning that the two human

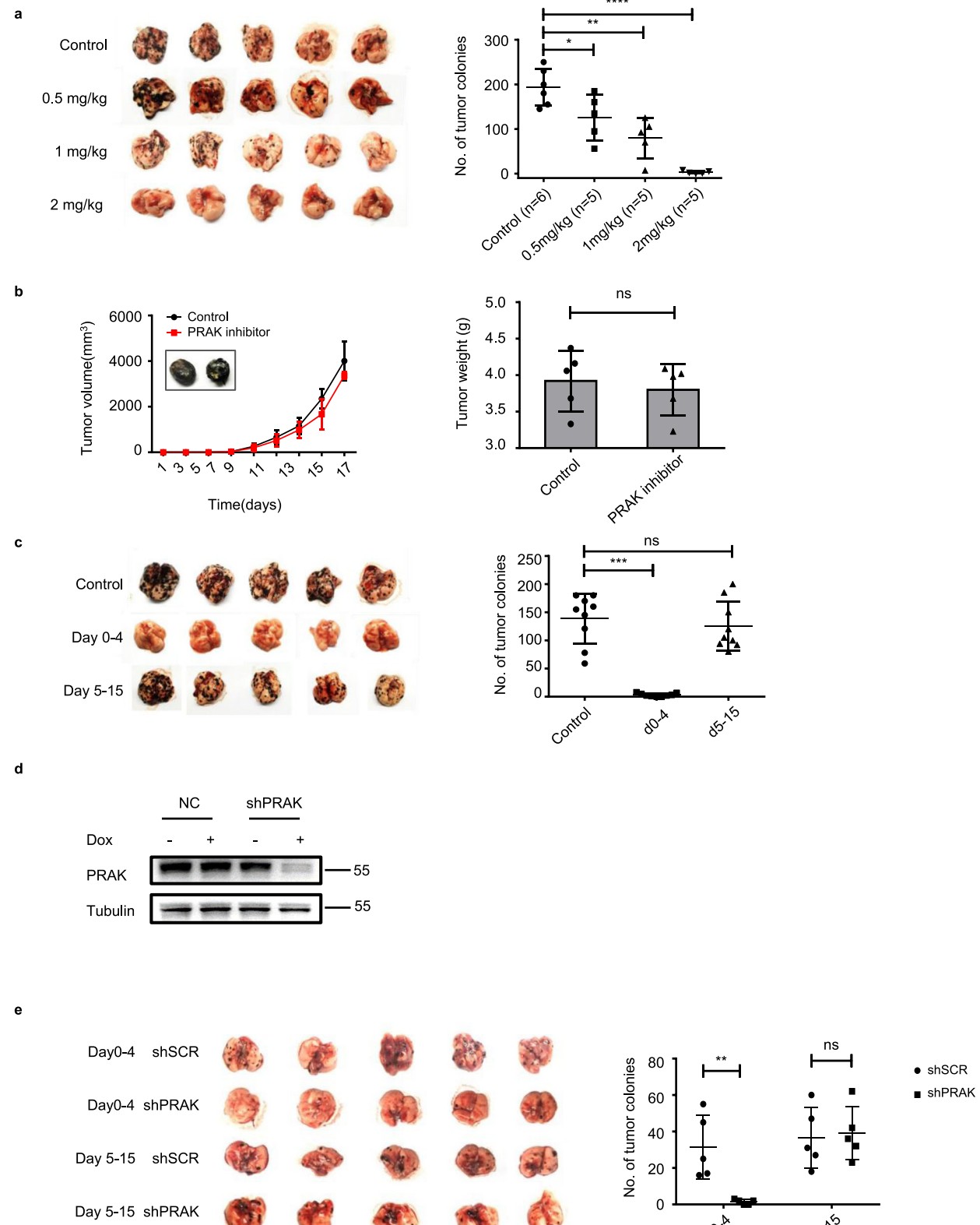

tumor lines were tested in immunodeficient mice, indicating that the immune system is not required for the anti-metastatic effect of PRAK inhibition. The effect of GLPG0259 was clearly dose-dependent, with virtually complete inhibition at a daily dose of 2 mg/kg (Fig. 2a). The growth of subcutaneously inoculated B16 cells, however, was not affected even with the highest dose of

GLPG0259 (Fig. 2b). Notably, the PRAK inhibitor displayed a critical time window of action. While its delivery on the first five days after tumor cell injection gave rise to similar results to full course treatment, inhibition was hardly detected if the drug was administrated after day 5 (Fig. 2c). To further test the time window at the genetic level, a B16 subclone expressing shPRAK

under the control of a Tet-On promoter was created (Fig. 2d) and used in the lung metastasis model. Consistent with the results obtained with the PRAK inhibitor, inducible knockdown in the first 5 days markedly inhibited lung metastasis, whereas delayed administration of Doxycycline in the last 10 days had virtually no effect (Fig. 2e). Taken together, these findings suggest that targeting PRAK holds the promise for the intervention of tumor metastasis.

**PRAK enhances invasion and colonization of tumor cells.** We next analyzed the cellular basis for the impaired metastasis in the absence of PRAK. *Prak* knockout B16 mutants were virtually identical to the parent cells in proliferation as measured by MTS assay (Supplementary Fig. 2a) and EdU incorporation (Supplementary Fig. 2c), either in normoxic or hypoxic cultures (Supplementary Fig. 2c). Comparable cell proliferation was also observed for B16 cells in the presence or absence of GLPG0259 (Supplementary Fig. 2b and c), and for *PRAK* knockdown and control A375 and MDA-MB-231 cells (Supplementary Fig. 2a). Moreover, Annexin V/7AAD staining demonstrated that neither *Prak* depletion nor PRAK inhibition caused any change in cell survival (Supplementary Fig. 2d). It held true even in the presence of apoptosis-inducing agent cisplatin (Supplementary Fig. 2e). These results were in line with the unaltered growth of subcutaneously inoculated B16 cells with the loss of function of PRAK. On the other hand, deficiency or inhibition of PRAK resulted in a significant decrease in the capacity of migration (Fig. 3a) and invasion (Fig. 3b) of B16 cells. Similar results were obtained with A375 (Supplementary Fig. 2f), MDA-MB-231 (Supplementary Fig. 2g), and HCT116 (Supplementary Fig. 2h) cells. In support of the functional specificity of PRAK, pharmacological inhibition of MK2, which is closed related to PRAK[30], had no effect in the invasion assay (Fig. 3b).

Metastasis is a complex multi-step process. To elucidate the critical step targeted by PRAK intervention, we monitored the fate of firefly luciferase-expressing MDA-MB-231 cells at different time points after intravenous injection. As shown in Fig. 3c, a large number of the cells were accumulated in the lung soon after injection. The photon signal faded rapidly and reached its nadir within 5 days. There was no overt difference among the control, *PRAK* knockdown, and inhibitor-treated groups at the initial stage. In the control group, the photon signal recovered soon and reached a new high in 2–3 weeks. In comparison, it remained at relatively low levels in both knockdown and inhibitor-treated groups (Fig. 3c). These results were confirmed by H&E staining of the lung sections (Supplementary Fig. 3a). Of note, the few metastatic lesions formed in the knockdown and inhibitor-treated groups were similar or only slightly reduced in size in comparison to that in control groups (Supplementary Fig. 3a), suggesting that, once metastases were established, PRAK had no or minimal effect on their outgrowth. To further dissect the early events in metastasis, *Prak*[+/+] and *Prak*[−/−] B16 cells were labeled with different dyes, mixed at an equal ratio, and intravenously injected into the recipient mouse. At different time points, intravascular perfusion staining was performed with tomato lectin to label the capillary endothelium, followed by PBS perfusion to remove free cells in the blood vessels. The two-photon microscope revealed that *Prak*[+/+] and *Prak*[−/−] B16 cells were equally present in the lung parenchyma 6 h after injection (Fig. 3d), indicating that *Prak*-deficiency did not interfere with the extravasation of tumor cells. Thereafter, we saw a marked decrease in both *Prak*[+/+] and *Prak*[−/−] cells. Nevertheless, the *Prak*[−/−] cells were apparently more vulnerable. By day 4, there was hardly any *Prak*[−/−] cells left (Supplementary Fig. 3b). Similar results were obtained when dyes were reversed, excluding the possibility of dye-related toxicity

(data not shown). Collectively, these results suggest that loss of function of PRAK primarily affects the colonization of metastatic cells in the distant organs.

**HIF-1α is critically involved in the regulation of tumor metastasis by PRAK.** In order to gain an insight into the molecular mechanisms behind the pro-metastatic activity of PRAK, RNA sequencing was employed to compare the transcriptional profiles of parent and *Prak* knockout B16 cells. More than 2400 genes were found to be differentially expressed (Fig. 4a). Gene ontology (GO) analysis revealed significant enrichment of genes in multicellular organism development, glutathione metabolic process, and response to hypoxia (Fig. 4b). Gene set enrichment analysis (GSEA) further revealed the coordinated downregulation of gene sets related to the hypoxic response and mTOR signaling (Fig. 4c). In view of the master regulatory role of hypoxia-inducible factor 1α (HIF-1α) in metastasis, our analyses were subsequently focused on the HIF-1α pathway. Normally, the HIF-1α level was markedly elevated under hypoxic conditions. Its induction, however, was attenuated in *Prak* knockout or inhibitor-treated cells (Fig. 4d). Consistent with this result, HIF-1α expression was profoundly suppressed in tumor cells isolated from *Prak*[−/−] PyMT mice versus those from *Prak*[+/+] PyMT mice (Fig. 4e). More intriguingly, both PRAK and HIF-1α were found to be increasingly expressed in the lung metastatic lesions in comparison to the primary tumors (Fig. 4f), implying a bias towards higher expression of PRAK and HIF-1α in tumor metastasis. To verify the role of HIF-1α in PRAK-mediated tumor metastasis, *PRAK* knockdown MDA-MB-231 cells were transfected with HIF-1α expression constructs. While the enforced expression of HIF-1α had no significant impact on the invasion of control cells, it rescued, to a large extent, the impaired invasive capacity of the *PRAK* knockdown cells. Not surprisingly, the rescuing effect was dependent on an intact transcription activation domain (TAD) of HIF-1α (Fig. 4g). In agreement with the in vitro data, overexpression of HIF-1α but not HIF-1αΔTAD restored lung metastasis by *PRAK* knockdown MDA-MB-231 cells (Fig. 4h).

Multiple metastasis-related genes are subjected to transcriptional regulation by HIF-1α[5]. Along with the decreased expression of HIF-1α, *Prak*-depleted or inhibitor-treated cells showed a significant reduction in MMP2 expression, whereas MMP9, another downstream target, remained unaltered (Supplementary Fig. 4a). In addition, the inactivation of PRAK was accompanied by the upregulation of E-cadherin but downregulation of N-cadherin, indicative of a suppressed EMT (Supplementary Fig. 4b). However, we saw no consistent changes in other mesenchymal markers like vimentin, Snail1, and Twist1 (Supplementary Fig. 4b). This may be partly explained by the recent findings that EMT is rarely fully activated in tumor cells. Instead, the transition is often accompanied by the loss or acquisition of only a subset of epithelial or mesenchymal traits[31]. To elucidate the role of HIF-1α in the altered expression of MMP2 and EMT-related molecules, HIF-1α was overexpressed in *PRAK* knockdown MDB-MB231 cells. Indeed, the enforced expression of HIF-1α rescued the distorted expression pattern induced by PRAK knockdown (Supplementary Fig. 4c and d).

**PRAK promotes HIF-1α translation by regulating mTORC1 activity.** HIF-1α expression is regulated at several levels, including transcription, translation, and degradation[5]. Real-time quantitative PCR demonstrated that PRAK deletion or inhibition had no impact on *Prak* mRNA levels (Fig. 5a). We next treated cells with the proteasome inhibitor MG132 to determine whether PRAK inactivation affected HIF-1α stability. While MG132 led to an overall

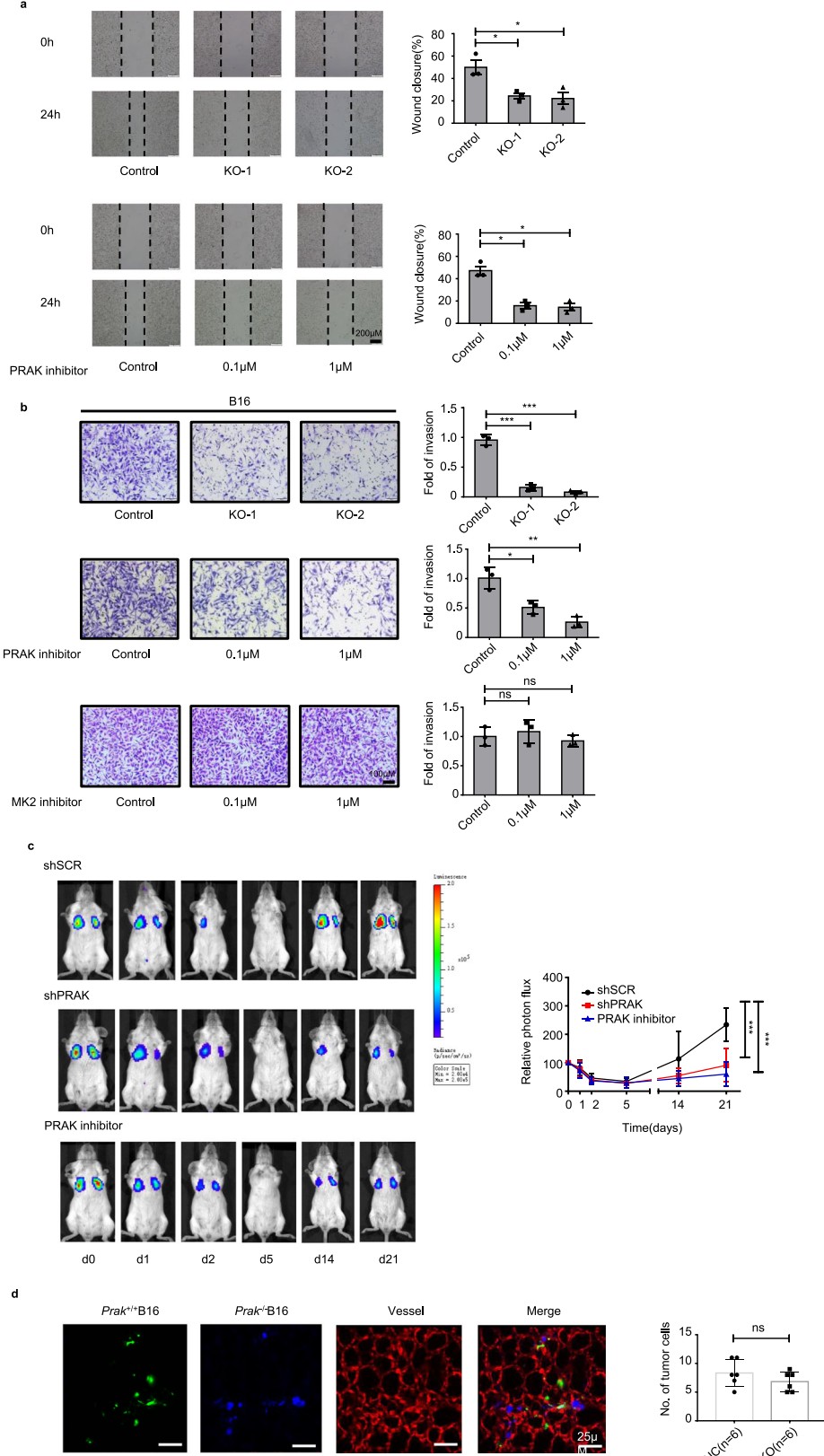

increase of the HIF-1α protein level, the difference between the control and *Prak*-deficient or inhibitor-treated cells was maintained (Fig. 5b), denying the possibility of altered protein degradation. In further support of this notion, the ubiquitination of HIF-1α was comparable with or without functional PRAK (Fig. 5c). Furthermore, we measured the protein synthesis of HIF-1α. Cells were pulsed with [$^{35}$S]-L-methionine and [$^{35}$S]-L-cysteine. HIF-1α was immunoprecipitated from the cell lysate, resolved by SDS-PAGE, and detected by autoradiography. As shown in Fig. 5d, loss of function of PRAK resulted in a drastic reduction in the synthesis of nascent HIF-1α proteins. Therefore, PRAK controls HIF-1α expression primarily through translational regulation.

**Fig. 3 PRAK promotes tumor cell invasion in vitro and colonization in vivo. a** Wound healing assay was performed to compare the migration of *Prak*$^{+/+}$ versus *Prak*$^{-/-}$ B16 clones (top) or *Prak*$^{+/+}$ cells in the presence or absence of the PRAK inhibitor GLPG0259 (bottom). The experiments were repeated three times with triplicates. Representative images and quantification of wound closure are shown. Error bars represent s.e. Scale bars, 200 μm. *$p < 0.05$. **b** Cell invasion was compared using Matrigel assay between *Prak*$^{+/+}$ and *Prak*$^{-/-}$ B16 clones (top), and *Prak*$^{+/+}$ cells in the absence or presence of PRAK inhibitor (middle) or MK2 inhibitor PF3644022 (bottom). The experiments were repeated at least three times. Representative images and quantification of fold of invasion are shown. Error bars represent s.d. Scale bars, 100 μm. ns, no significant, *$p < 0.05$, **$p < 0.01$, ***$p < 0.001$. **c** MDA-MB-231-Luc-D3H2LN cells carrying shPRAK or control scrambled shRNA (shSCR) were intravenously injected into SCID mice. The PRAK inhibitor-treated group received daily intraperitoneal injection of inhibitor at 2 mg/kg from day 0 to 4. Lung metastasis was monitored by bioluminescent imaging at different time points. Representative images are shown on the left. Data collected from five mice in each group are presented as mean ± s.d. ***$p < 0.001$. **d** Equal numbers of *Prak*$^{+/+}$ (green) and *Prak*$^{-/-}$ (blue) B16 cells were mixed and injected into the mouse through the tail vein. Colonization of tumor cells in the pulmonary parenchyma was examined using confocal microscopy after perfusion. Representative images are shown at hour 6 post-injection. The blood vessels were stained red. The experiments were repeated at least three times with similar results. *p*-value was determined by a two-tailed, unpaired *t*-test.

As revealed by the transcriptome analysis, downregulation of the mTOR signaling-related gene set was one of the most prominent changes associated with *Prak* deficiency (Fig. 4c). Notably, mTORC1 is known to be important for the cap-mediated translation of a specific subset of transcripts bearing 5′ TOP, including HIF-1α mRNA[32,33]. Therefore, we set to examine how mTOR signaling was affected by PRAK. As shown in Fig. 5e, the phosphorylation of mTORC1 and its downstream substrate S6K1 was significantly decreased in *Prak*-deleted or GLPG0259-treated B16 cells. To ascertain the role of mTORC1 in PRAK-regulated HIF-1α translation and cell invasion, *Prak* knockout B16 cells were treated with the mTORC1 agonist MHY1485. Indeed, MHY1485 treatment increased the expression of HIF-1α and its target protein MMP2 to a level comparable to that in *Prak*-sufficient cells (Fig. 5f), along with the restored expression of N-cadherin and E-cadherin (Supplementary Fig. 5a). Moreover, the impaired invasion of the knockout cells was partially rescued (Fig. 5g). On the other hand, the mTORC1 inhibitor rapamycin, similar to the PRAK inhibitor, imposed a suppressive effect on the invasion of B16 cells (Supplementary Fig. 5b) as well as the lung metastasis of ER$^+$ breast cancer cell MCF7 (Supplementary Fig. 5c). Taken together, these data support that PRAK plays an important role in the translational regulation of HIF-1α, which is at least in part mediated by mTORC1.

**PRAK expression correlates with metastasis and survival in human cancer.** To evaluate the pathological significance of the pro-metastatic activity of PRAK in human cancer, we first analyzed PRAK expression in a cohort of 60-non-small cell lung cancer (NSCLC) samples, which were subgrouped according to whether distant metastasis occurred within 5 years after surgical removal of the primary tumor. Quantitative RT-PCR demonstrated that *Prak* mRNA expression was significantly higher in the 30 samples from patients with the late occurrence of metastasis than the subset ($n = 30$) with no metastasis (Fig. 6a). We next assessed PRAK and HIF-1α protein expression using immunohistochemistry in a human tissue array containing 40 lung carcinoma samples with paired lymphatic metastases. Consistent with many other previous reports[6,34–36], HIF-1α showed more intensive staining in metastatic than primary tumors (Fig. 6b and d). Notably, the paired lymphatic metastasis also showed significantly higher H-scores for PRAK (Fig. 6b and c). Moreover, the expression levels of PRAK strongly correlated with those of HIF-1α (Fig. 6e). Similar results were obtained in another tissue array composed of breast cancer paired with lymphatic metastasis (Supplementary Fig. 6a–d). We also explored the potential link between PRAK expression and patient survival using a publically accessible online tool KM Plotter (http://kmplot.com). The 1926 lung cancer patients were divided into two groups of similar size based on the median value of PRAK expression. Kaplan–Meier survival analysis indicated that higher levels of PRAK were associated with reduced overall survival in lung cancer patients (Fig. 6f, $p = 1.7 \times 10^{-8}$). Similar analyses were conducted with a cohort of 536 breast cancer patients and another cohort of 230 skin melanoma patients A significant correlation was observed for breast cancer (Fig. S7a, $p = 0.036$) but not for melanoma patients (Fig. S7b, $p = 0.94$). Together, these data suggest that elevated PRAK expression may be predictive of increased incidence of metastasis and poor survival in human lung and breast cancer.

**Discussion**

A complex role has previously been reported for PRAK in tumorigenesis. On the one hand, it suppresses tumor development by promoting cell senescence and growth arrest[17,19,23]. On the other hand, it accelerates the progression of established tumors by stimulating angiogenesis[21]. Here we demonstrate that PRAK plays a unique role in tumor metastasis. In PyMT mice, *Prak* deficiency abrogated the lung metastases of breast cancer, whereas the incidence and growth of the primary tumors remained unaltered. Similar results were obtained in tumor transplantation models with several murine and human tumor lines. In each case, *Prak* deletion or knockdown, while imposing no impact on the in situ growth of subcutaneously inoculated tumor cells, resulted in a much-suppressed formation of metastatic foci in the lung tissue following intravenous injection. Consistent with a pro-metastatic role of PRAK, we detected higher levels of PRAK in metastatic lesions than primary tumors in PyMT mice. Increased PRAK expression was also observed in lymphatic metastases in comparison to the paired human lung or breast cancer samples. Moreover, there was a significant inverse correlation between the PRAK level and overall survival in lung cancer patients.

The pro-metastatic activity of PRAK is in contrast to the anti-metastatic function of MSK1, another MAPK-activated protein kinase downstream of p38MAPK[37]. In that study, Gawrzak et al. have demonstrated that p38MAPK activates MSK1, which in turn strengthens metastatic latency in ER$^+$ breast cancer by promoting luminal cell differentiation. Their results render further support for an important role of p38MAPK signaling in the maintenance of metastatic tumor dormancy. However, a pro-metastatic effect of p38MAPK has also been reported in a wide array of tumors[38–40]. Therefore, the exact function of p38 signaling in tumor metastasis appears to be highly dependent on the specific signaling cascade initiated downstream of p38MAPK.

By activating the transcription of a wide array of genes involved in various steps of the metastatic cascade, HIF is centrally positioned in the regulatory network governing tumor metastasis[6,7]. In the absence of functional PRAK, the protein level of HIF-1α was found to be markedly reduced. In support of a key role of HIF-1α in the pro-metastatic activity of PRAK, enforced expression of HIF-1α restored to a large extent the impaired invasion and lung colonization of *Prak* knockdown MDA-MB-231 cells. Further analyses indicated that PRAK deficiency had no impact on HIF-1α mRNA expression or protein degradation but

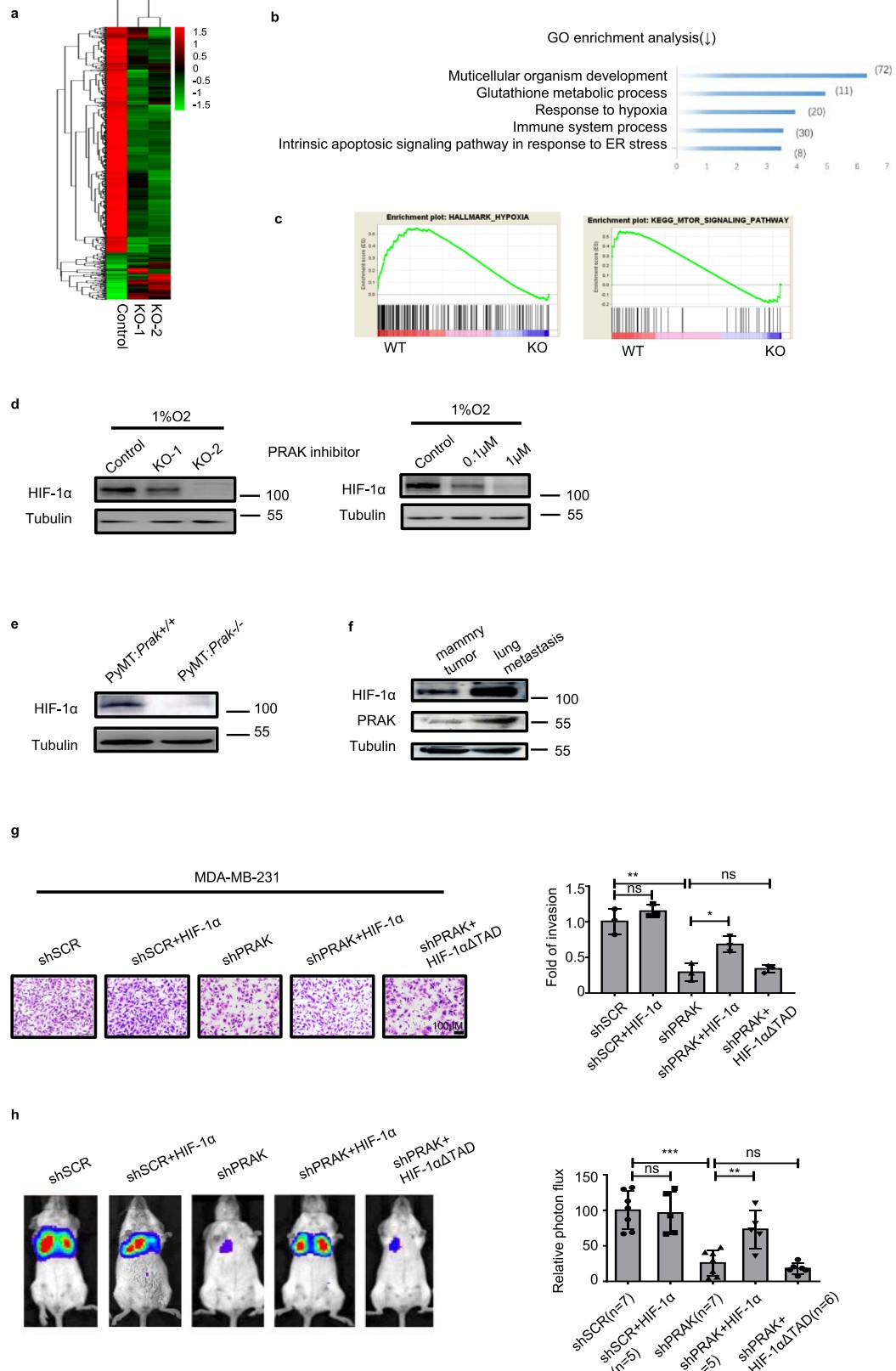

caused a profound inhibition of protein synthesis. Translational regulation represents an important mechanism for the control of intracellular levels of HIF-1α. Several RNA-binding proteins, including YB-1[41], HuR, and PTB[42], bind to HIF-1α mRNA and promote its translation. Besides, mTORC1 selectively enhances the cap-mediated translation of a subset of mRNAs bearing 5'TOP, including HIF-1α mRNA[32,33]. PRAK deletion or inhibition was noted to be associated with decreased phosphorylation of mTORC1 and its substrate S6K1. Treatment with the mTORC1 agonist MHY1485 led to partial recovery of the reduced HIF-1α expression and invasiveness of *Prak*-deleted tumor cells. These data suggest that the pro-metastatic activity of

**Fig. 4 The metastasis-promoting effect of PRAK is mediated by HIF-1α. a** The transcriptional profiles of parent and *Prak* knockout B16 clones were determined by RNA-Seq. The heatmap shows the clustering of the differentially expressed genes. **b** The top five biological functional pathways revealed by GO analysis of the differentially expressed genes. **c** The coordinated downregulation of gene sets related to hypoxia (NES = 1.4548125, NOM *p*-val = 0.0) and mTOR signaling (NES = 1.12915, NOM *p*-val = 0.0) in *Prak*−/− cells as revealed by GSEA analysis. **d** Western blotting analysis of HIF-1α levels in *Prak*+/+ and *Prak*−/− B16 cells cultured in 1%O$_2$ for 12 h in the presence or absence of GLPG0259. The experiments were repeated 3 times with similar results. **e** HIF-1α levels in mammary tumor tissues isolated from *Prak*+/+ and *Prak*−/− PyMT mice. **f** PRAK and HIF-1α levels in mammary tumors in comparison to metastatic lesions in the lung isolated from the same PyMT mouse. **g** MDA-MB-231-Luc-D3H2LNc cells were infected with lentiviruses carrying shSCR, shSCR+HIF-1α, shPRAK, shPRAK+HIF-1α, or shPRAK+ HIF-1αΔTAD. The invasive capacity of these cells was analyzed using Matrigel assay. The experiments were repeated three times. Representative images and quantification of fold of invasion are shown. Error bars represent s.d. Scale bars, 100 μm. *\*p* < 0.05, *\*\*p* < 0.01, ns, not significant. **h** The MDA-MB-231-Luc-D3H2LN cells described above were tested for the formation of lung metastasis following intravenous injection. Bioluminescence was recorded on day 21. Each group contained 5–7 mice. Representative images and relative photon flux are shown. Error bars represent s.d. *\*\*p* < 0.01, *\*\*\*p* < 0.001, ns, no significant. *p*-value was determined by a two-tailed, unpaired *t*-test.

PRAK is at least partially mediated by enhanced mTORC1 activation and HIF-1α translation. Much work, however, is required to determine the molecular mechanism for PRAK-induced activation of mTORC1 in tumor cells. In fact, it is contradictory to a previous report by Zheng et al.[18], in which PRAK was shown to act as a negative regulator of mTORC1 under energy depletion. Thus, PRAK may exert opposite effects on mTORC1 activation under different conditions in different cell types.

It has been suggested that the post-extravasation regulation of tumor growth is most critical in determining metastatic outcome[5]. By tracing the fate of fluorescence-labeled tumor cells, we demonstrated that the initial infiltration of *Prak*−/− tumor cells into the lung tissue was comparable to that of the wild-type cells after intravenous injection. But most of the *Prak*−/− cells were eliminated in the next few days, indicating that PRAK mainly functions at the late stage of the metastatic cascade. In vitro studies demonstrated that PRAK inactivation was accompanied by impaired cell migration and invasion, which presumably could be important for tumor cells to locate a pro-metastatic niche. The role of PRAK in cell motility has been previously addressed by several studies, but with contradictory results. In one study, Tak et al.[43] reported that PRAK promoted cell motility by phosphorylating HSP27. In contrast, the study by Stohr et al.[44] suggested that PRAK-mediated HSP27 phosphorylation reduced the velocity of tumor cell migration. Regardless of these findings, PRAK-mediated HSP27 phosphorylation is unlikely responsible for the impaired migration observed here. Possibly due to the functional redundancy of other members of this family including MK2 and MK3[45], no difference in HSP27 phosphorylation was seen with the loss of PRAK function (Supplementary Fig. 8a). On the other hand, PRAK inactivation was found to be associated with the upregulation of E-cadherin and concomitant downregulation of N-cadherin, indicative of suppressed EMT. As the contribution of EMT to tumor invasion and the role of HIF-1α in the transition are well established, it is reasonable to speculate that the altered PRAK-HIF-1α-EMT axis may participate in the inhibition of cell invasion. In support of this speculation, enforced HIF-1α expression restored the invasiveness of *Prak*-deficient tumor cells. Successful metastatic colonization also relies on the capacity of the infiltrating tumor cells to adapt to the metabolic stress frequently experienced in a new microenvironment and on angiogenesis to support the growth of macroscopic metastases[5]. Like EMT, both metabolic reprogramming and neoangiogenesis are orchestrated by HIF-1α[3,6]. Therefore, the decreased level of HIF-1α induced by PRAK inactivation could affect multiple aspects of metastatic cells, each of which may contribute to the inhibition of tumor colonization. However, it should be pointed out that, although lung metastasis in PyMT mice was similarly impaired with the loss of function of either HIF-1α or PRAK, the defect was apparently less severe in the absence of HIF-1α[46], indicating that additional mechanisms

may contribute to the PRAK-mediated regulation of tumor metastasis.

What we reported here has important clinical implications. Despite the recent advancements in cancer therapy including immunotherapy[47], a metastatic disease largely remains incurable. To improve the long-term outcomes of cancer patients, many current studies are attempting to inhibit various steps of the metastatic process, from reducing the emergence of invasive tumor cells in primary tumors to suppressing the growth of disseminated tumor cells at the secondary sites[48]. MMPs, for example, have long been heralded as potential targets of intervention on the basis of their general upregulation in virtually all human and animal malignancies and their key roles in extracellular matrix remodeling. While profound inhibitory effects were repeatedly observed on the invasive behavior of tumor cells in experimental animal models, the results of human clinical trials with MMP inhibitors are disappointing[49]. Since metastatic cell growth at distant sites requires the development of a vascular supply, targeting VEGF or its receptors has thus attracted much attention. However, the clinical efficacy of this approach was also limited. In some cases, anti-VEGF therapy was even shown to accelerate the formation of metastasis[50]. The difficulties in targeting metastasis may be partly due to the multiple parallel mechanisms of dissemination[3]. Identification of common bottlenecks will therefore be critical toward the goal of developing metastases-specific therapies. Given its diverse and potent roles in tumor progression and metastasis[6,7,10], HIF may represent one of such targets. Indeed, searching for therapies to inhibit HIF activity has been the focus of intense study in recent years[51]. However, it is technically challenging to target transcription factors in general. It is therefore not surprising that there is currently no inhibitor selectively and directly targeting HIF. Alternatively, the HIF signaling may be modulated by targeting up or down stream components. The present study identifies PRAK as a promising candidate for such a purpose. Its inhibitor GLPG0259 not only induced a significant reduction in HIF-1α expression, but also fully mimicked the metastasis-suppressive effect generated by *Prak* depletion. A phase II trial had been performed for GLPG0259 in the treatment of rheumatoid arthritis. Although the trial was terminated due to the lack of efficacy, it provides evidence that the compound possesses excellent pharmacokinetic properties and tolerability[28,29].

An obvious scenario to which GLPG0259 could be applied is the prevention of tumor spreading during the perioperative period. Surgical resection remains the primary choice in the cure of most solid cancers. Paradoxically, surgical insult often leads to a remarkable rise in circulating tumor cells (CTC). Moreover, there is a strong correlation between tumor recurrence and CTC levels before and during surgery[52,53]. Therefore, the perioperative period might provide a unique window for the control of metastasis[54,55]. Several studies have highlighted the beneficial

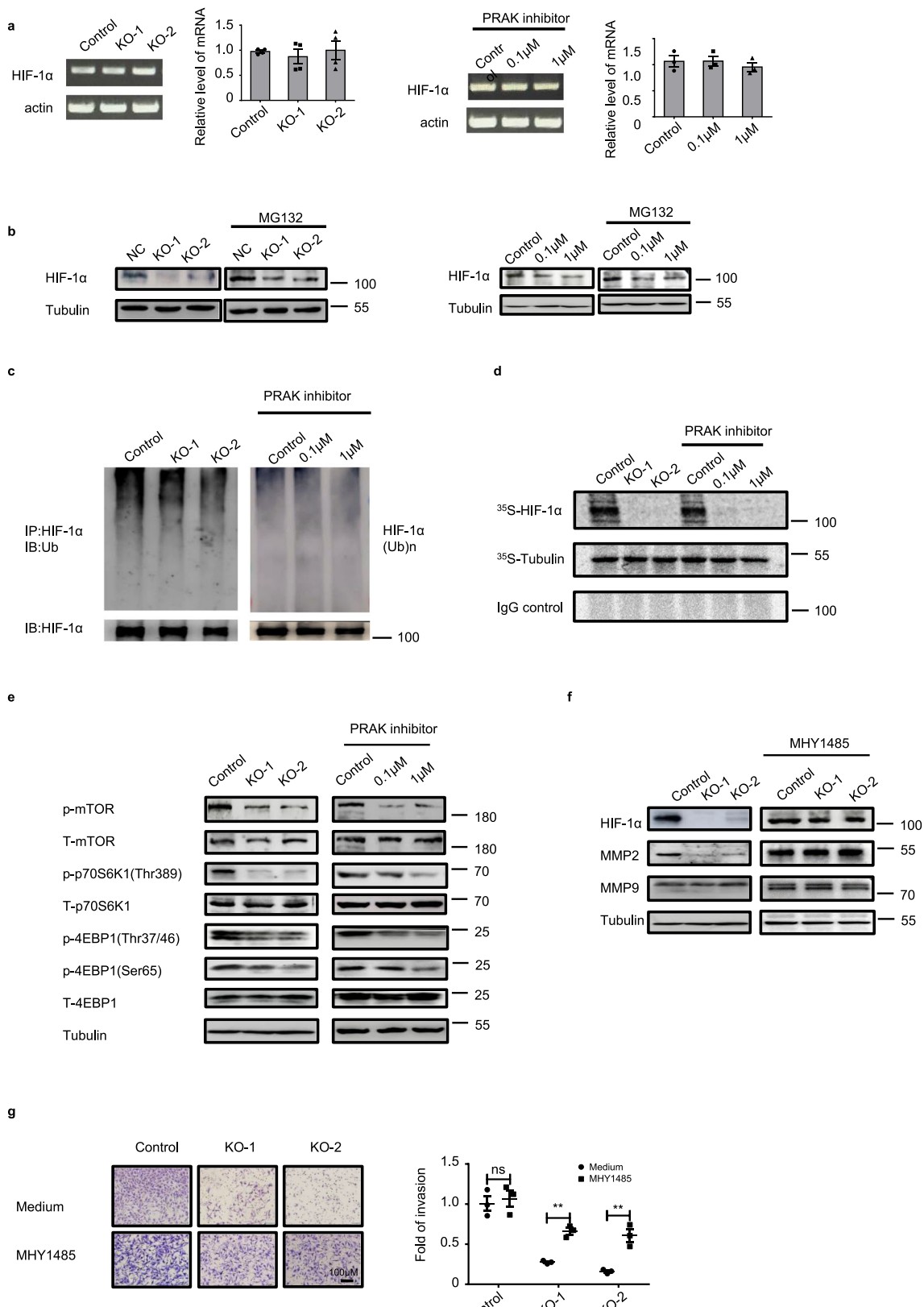

effect of perioperative treatment with COX inhibitors or beta blockers, but the results are inconclusive[54]. Given the potent anti-metastatic effect of the PRAK inhibitor in animal models, it is worthy being tested in clinical settings.

In summary, both genetic and pharmacological evidence points to a potent pro-metastatic activity of PRAK, which is partly mediated by enhanced HIF-1α translation. Targeting PRAK holds promise in the intervention of tumor metastasis.

**Fig. 5 PRAK promotes HIF-1α translation by regulating mTORC1 activity. a** HIF-1α mRNA levels in *Prak*$^{+/+}$ and *Prak*$^{-/-}$ B16 cells (left) and in B16 cells treated with the PRAK inhibitor (right) as measured by real-time RT-PCR. An agarose gel of the PCR products is shown on the Left. **b** HIF-1α protein levels in parent, *Prak*$^{-/-}$ and GLPG0259-treated B16 cells in the absence or presence of MG132 as determined by Western blotting. Tubulin served as a loading control. **c** B16 cells were cultured in the presence of MG132 for 10 h. HIF-1α was precipitated from the lysate and the blot was probed with anti-ubiquitin antibodies. **d** B16 cells were cultured in the presence of CoCl$_2$ with the addition of 500μCi of [$^{35}$S]-L-methionine and [$^{35}$S]-L-cysteine for 6 h. Immunoprecipitation was performed with antibodies against HIF-1α or tubulin or with control IgG. Newly synthesized proteins of the expected size were detected by autoradiography. **e** Cell lysate was prepared from control, *Prak* knockout, and inhibitor-treated B16 cells. The blot was probed with anti-phospho- or total mTOR, S6K1 and 4EBP1. Tubulin served as a loading control. **f, g** *Prak*$^{+/+}$ and *Prak*$^{-/-}$ B16 cells were cultured with or without the mTORC1 agonist MHY1485 (1 μM for 12 h). HIF-1α, MMP2, and MMP9 expression was detected by immunoblotting (**f**). The invasive capacity of these cells was analyzed using Matrigel assay (**g**). Each experiment was repeated at least three times with similar results. ns no significant, **$p < 0.01$. $p$-value was determined by a two-tailed, unpaired $t$-test.

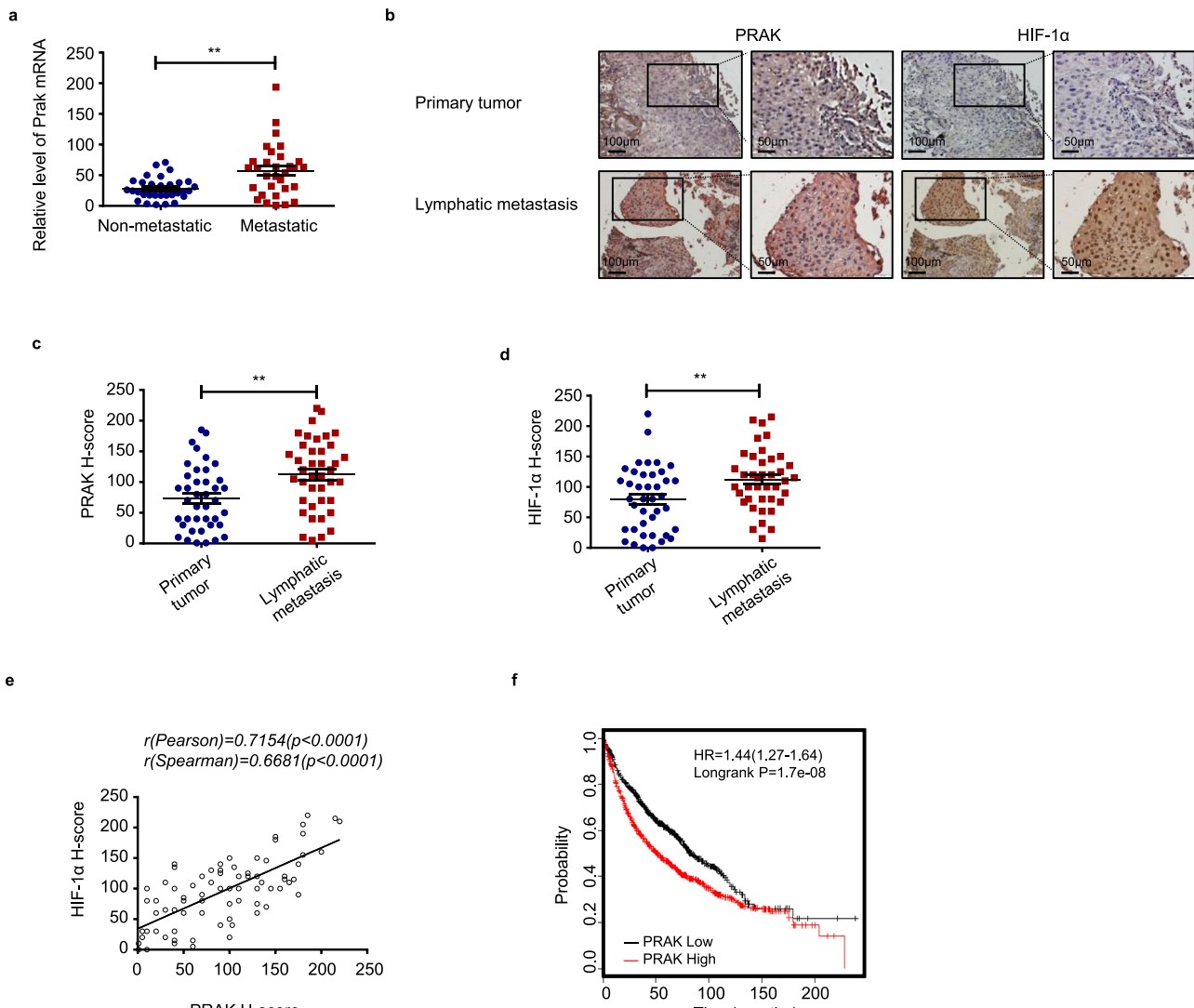

**Fig. 6 PRAK expression is elevated in metastatic lesions of human cancer. a** *Prak* mRNA expression was quantified by real-time RT-PCR in 60 lung carcinoma samples with or without recurrence over a 5-year follow-up (30 for each group). Each symbol represents an individual sample. Bars are the mean and error bars represent s.e. **$p < 0.01$. **b–e** A human tissue array containing 40 lung carcinoma samples with paired lymphatic metastasis was analyzed for PRAK and HIF-1α expression by immunohistochemistry. Representative images (**b**) and H-scores for PRAK (**c**) and HIF-1α (**d**) staining are shown. Each symbol represents an individual sample. Bars are the mean and error bars represent s.e. **$p < 0.01$. The correlation of PRAK and HIF-1α expression was analyzed using R programming (**e**). **f** Kaplan–Meier survival analysis was performed with a cohort of 1926 lung cancer patients from a public database (http://kmplot.com/analysis/) to explore the correlation of survival time with PRAK expression. $p$-value was determined by two-tailed, unpaired $t$-test (**a**) and paired $t$ test followed by Wilcoxon test (**c, d**).

## Methods

**Mice.** C57BL/6 and SCID animals were obtained from Peking University Health Science Center or Vital River Lab Animal Technology Company (Beijing, China). MMTV-PyMT mice were a gift from Professor Xiaoren Zhang (Shanghai Institute for biological sciences, CAS). *Prak*-deficient mice were generated as previously described[21] and backcrossed onto the C57BL/6J background for 6-8 generations. *Prak*$^{+/-}$ mice were bred with MMTV-PyMT to generate *Prak*$^{+/+}$ MMTV-PyMT and *Prak*$^{-/-}$ MMTV-PyMT littermate mice. The animals were kept in a specific pathogen-free facility at Peking University Health Science Center. All the animal procedures were conformed to the Chinese Council on Animal Care Guidelines and the study was approved by the ethics committee of Peking University Health Science Center with an approval number of LA2014178.

**Antibodies and reagents.** The following antibodies were used for Western blot analyses, immunoprecipitation, and immunohistochemistry: anti-mTOR, anti-phospho-mTOR$^{Thr2448}$, anti-p70S6K, anti-phospho-p70S6K$^{T389}$, anti-4EBP1, anti-phospho-4EBP1$^{Thr37/46}$, anti-phospho-4EBP1$^{Ser65}$ and anti-phospho-HSP27 from Cell Signaling; anti-PRAK, anti-HIF-1α and anti-MMP2 from Proteintech (Chicago, IL), anti-tubulin and anti-GAPDH from Ruiying Biological (Suzhou, China), anti-vimentin from Gene Tex (San Antonio, Texas), anti-Snail1, anti-MMP9, anti-CDH1(E-cadherin) and anti-CDH2 (N-cadherin) from ABclonal (Wuhan, China), anti-Twist1 from Signalway Antibody (College Park, Maryland), Isotype control IgG from Sigma. PRAK inhibitor GLPG0259 was synthesized by Wuxi Pharma (Shanghai, China). MK2 inhibitor PF 3644022 was purchased from R&D systems. mTORC1 inhibitor rapamycin and mTORC1 agonist MHY1485 were from Selleck (Houston, TX), Texas red-labeled tomato lectin was from Vector Laboratories (Burlingame, CA).

**Cell lines.** B16 murine melanoma cell, A375 human melanoma cell and MCF7 human breast carcinoma cell were obtained from ATCC and maintained in the appropriate medium with supplements as suggested. MDA-MB-231-Luc-D3H2LN human breast carcinoma cells stably expressing firefly luciferase were a kind gift from Professor Yongfeng Shang (Peking University) and were cultured in L-15 medium supplemented with 10% FBS and without $CO_2$ at 37 °C. HCT116 human colon cancer cells were obtained from ATCC and maintained in McCoy's 5A (modified) Medium supplemented with 10% FBS. Hypoxic cultures were set up in 1–5% $O_2$. *Prak* knockout B16 cells were generated using the CRISPR/Cas9 technology. Briefly, optimal guide RNAs specific for *Prak* were selected for ligation with the px458 plasmid. Cells were transfected with the construct and subjected to monoclonal culture. Multiple clones were obtained. *Prak*-deficiency was verified by DNA sequencing and Western blotting. Two clones were used throughout the study. *Prak* knockdown A375 and MDA-MB-231 cells were prepared through infection with recombinant lentiviruses expressing shPRAK (shPRAK-1: GAAATTGTGAAGCAGGTGATA, or shPRAK-2: CCAAAGGACAGTGTCTATATC) or control scrambled shRNA (shSCR: TTCTCCGAACGTGTCACGTAA). *Prak* inducible knockdown B16 cells were created by transfection with the HBLV lentiviral vector expressing shPRAK (GCACTGTCACTTGCTAAACAT) under the control of a Tet-On promoter. For the induction of PRAK knockdown, 2 mg/ml doxycycline was supplied in the drinking water 1 day before or 4 days after tumor cell inoculation. Lentiviral vectors were also used to express exogenous HIF-1α and HIF-1αΔTAD in MDA-MB-231. All recombinant lentiviruses were purchased from HanBio Technology (Shanghai, China).

**Tumor specimens.** The present study has been approved by the Institution Ethics Committee of Peking University (EAEC 2018-11). A total of 60 samples were collected from the I~III A stage NSCLC patients who have received surgery in Peking University Cancer Hospital with informed consent. Samples were frozen in liquid nitrogen immediately after surgical removal and maintained at −80 °C until mRNA extraction.

**Cell proliferation assay.** Cells were seeded into 96-well plates ($5 \times 10^3$/well). GLPG0259 was added to a final concentration of 0.1 or 1 μM when applicable. Triplicates were set up for each condition. At different time points, MTS (Promega, USA) was added (10 μl per 100 μl medium), and the absorbance at 490 nm (OD value) was recorded. In some experiments, cell proliferation was also assessed by EdU incorporation as instructed by the manufacturer (RiBobio, China).

**Cell apoptosis assay.** Cells were seeded into a 24-well plate ($5 \times 10^4$/well). GLPG0259 was added to a final concentration of 0.1 or 1 μM when applicable. Triplicates were set up for each condition. Cells were harvested 12 h later, washed, and resuspended in binding buffer (Sungene, Tianjin, China). Cells were sequentially incubated with Annexin V-APC for 20 min at 4 °C and 7-AAD for 5 min at room temperature in darkness before being analyzed with a flow cytometer (Galios, Beckman Coulter). In some experiments, cisplatin was added to a final concentration of 20 μM 24 h before harvest to induce cell apoptosis.

**Wound healing and matrigel invasion assay.** Cells were seeded into a 12-well plates at a density of $1 \times 10^5$/well with or without the addition of GLPG0259.

Triplicates were set up for each condition. Cells were allowed to adhere for 12 h before wounds were introduced using pipet tips. Wound closure was monitored 24 h afterwards. The gap distances were quantitatively evaluated using ImageJ. Percent closure was calculated as (initial distance-final distance)/initial distance. Matrigel invasion assay was performed with BioCoat Matrigel Invasion Chambers (BD Biosciences) according to the manufacturer's instructions. Fold of invasion was calculated as the average ratio of cells between different groups in 3–5 random field of view.

**Implantation of tumor cells and monitoring of tumor growth.** For subcutaneous inoculation, $5 \times 10^5$ B16 cells were injected at the right flank of 6–8-week-old female C57BL/6 mice. Tumor volume, which was calculated as (length × width$^2$)/2, was recorded at different time points. For intravenous implantation, $2 \times 10^5$ B16 cells and $1–2 \times 10^6$ MDA-MB-231, A375 or MCF7 cells were injected into the lateral tail vein of 6–8-week-old female C57BL/6 mice and SCID mice, respectively. Tumor colonies in the lung were counted at day 16 for B16, at day 29 for A375 cells and at day 30 for MCF7 cells. The tumors formed by MDA-MB-231 cells were monitored by bioluminescence imaging. Briefly, D-luciferin was intraperitoneally administrated at 200 mg/kg at various time points. Fifteen minutes after injection, mice were anesthetized and bioluminescence was imaged with a charge-coupled device camera (IVIS; Xenogen). MDA-MB-231 tumors in the lung were also examined by H&E staining, which was entrusted to Servicebio (Wuhan, China).

**Two-photon imaging.** *Prak*$^{+/+}$ and *Prak*$^{-/-}$ B16 cells were labeled with Cell-Tracker$^{TM}$ Red CMTPX, CellTracker$^{TM}$ Blue CMAC, or CellTracker$^{TM}$ Green CMFDA. Equal numbers of differently labeled cells were mixed and injected into the lateral tail vein of C57BL/6 mice. The mice were sacrificed 6, 24 or 96 h after injection and tumor cells in the lung tissue were examined using a confocal microscope (Leica TSC MP SP8) after perfusion with PBS. In some experiments, Texas Red® labeled Tomato Lectin were injected intravenously to mark the capillary endothelium 5–10 min before the mice were sacrificed.

**In vivo administration of the PRAK inhibitor GLPG0259 or Rapamycin.** GLPG059 was administrated through intraperitoneal injection at a dose of 0.5, 1, or 2 mg/kg. Rapamycin was used at a dose of 2 mg/kg. PyMT mice received GLPG0259 injection every two days between 14 and 16 weeks after birth. In tumor implantation models, a daily dose was given from day 0 to 4, day 5 to 15, or day 0 to 15.

**Reverse-transcription quantitative PCR.** Total RNA was extracted using TRIzol reagent (Invitrogen). Two microgram RNA was reverse transcribed into cDNA using a Reverse Transcription System kit (Promega, USA). Quantitative PCR was performed with the ECO real-time PCR instrument (Illumina) using a qPCR SYBR Green Supermix (Bio-Rad). The primers used here were synthesized by Beijing Genomics Institute (primer information in Supplementary Table 1). All experiments were repeated at least three times with triplicates for each sample.

**Western blot analysis.** Cell lysate were prepared by incubating with RIPA lysis buffer, size-fractionated by SDS-polyacrylamide gel electrophoresis (PAGE), and transferred onto intracellular membranes. The blots were incubated with appropriate antibodies overnight at 4 °C, followed by incubation with HRP-conjugated secondary antibodies. Immunoreactive bands were visualized using ECL western blotting substrate (Thermo Fisher) according to the manufacturer's recommendation.

**Ubiquitylation assays.** B16 cells were cultured with 20 μM MG132 for 10 h. The cell lysate was then incubated with anti-HIF-1α overnight at 4 °C with constant rotation, followed by incubation with protein A agarose beads for another 4 h. The precipitated protein was eluted from the beads by resuspending the beads in 2× SDS-PAGE loading buffer and boiling. After resolution by SDS-PAGE, HIF-1α ubiquitination was detected anti-ubiquitin antibodies (PTM biolabs Chicago,IL).

**[$^{35}$S] metabolic labeling.** B16 cells were cultured with 500 μCi of [$^{35}$S]-L-methionine and [$^{35}$S]-L-cysteine (PerkinElmer) for 6 h after pre-culturing in methionine-free and cysteine-free medium (Invitrogen). Cells were disrupted using a standard protein lysate protocol. Immunoprecipitation was performed with anti-HIF-1α, anti-tubulin, or isotype-matched control IgG. The precipitated proteins were resolved on a 10% SDS polyacrylamide gel. The gel was exposed to an X-ray film (PharoFX Plus) for 24 h.

**RNA-Seq and data analysis.** RNA-Seq and data analysis were performed as described[56]. Briefly, libraries were generated for B16 parent and *Prak* knockout clones with Illumina TruSeq RNA Sample Prep Kit (Illumina). Illumina Hiseq2500 was used for sequencing to produce double 150 bp paired-end reads. To estimate gene expression, we quantified transcript levels as reads per kb of exon model per million mapped reads (RPKM). The differentially expressed genes, defined as those with a RPKM ratio >2 between any two clones, were uploaded into the bioinformatics resource DAVID v6.7 (https://david.ncifcrf.gov/) to assess the enrichment of

input genes with GO biological process terms. GSEA was carried out by searching Molecular Signature Database (MSigDB) version 4.0 provided by the Broad Institute (http://www.broad.mit.edu/gsea/).

**Tumor tissue arrays and Immunohistochemistry**. Lung (LC817a) and breast (BR1005b) carcinoma tissue arrays were purchased from US Biomax (Rockville, MD). The expressions of PRAK and HIF-1α were analyzed by immunohistochemistry. Briefly, the paraffin-embedded tissue sections were deparaffinized and rehydrated. Antigen retrieval was performed in antigen retrieval buffers at 95 °C for 60 min. Endogenous peroxidase was blocked with 3% $H_2O_2$. After blocking with 5% normal goat serum (5%), sections were incubated with appropriate antibodies at 4 °C overnight, followed by adding dextran carrying anti-rabbit IgG conjugated to HRP. AEC and DAB two-color rendering methods were used to distinguish between two antibodies. A new round of antigen retrieval and subsequent steps were performed between the two stains to rule out the interference of the previous antibody. Development of the tissue array was carried out using the Motic Easy Scan detection system. Immunohistochemistry evaluations were performed by one pathologist and one trained reader who were blinded to the experimental data. The staining of PRAK and HIF-1α was scored (H score) by taking into account both the intensity of staining and the percentage of positive cells.

**Statistical analysis**. Results were reported as mean ± s.e. or mean ± s.d. for independent experiments with or without triplicates. Statistical analysis was performed with SPSS V.16.0 and GraphPad Prism 7. Two-tailed Student's $t$ tests were used for single comparison, and two-way ANOVA was used for multiple comparisons unless otherwise specified. The correlation coefficients were calculated by R programming. Data for Kaplan–Meier survival analysis of lung cancer patients were from http://kmplot.com/analysis/index.php?p=service&cancer=lung, whereas data for breast cancer and skin melanoma patient survival analyses were from http://gepia.cancer-pku.cn/.

## Data availability

The RNA-seq data have been made available at the NCBI Sequence Read Archive (SRA) repository under the accession number SRP300830. The KM plotter data that support the findings of this study are available from "Kaplan Meier-plotter", http://kmplot.com. The breast cancer and skin melanoma patient survival analysis data that support the findings of this study are available from "GEPIA", http://gepia.cancer-pku.cn/. The data that support the findings of this study are available from the corresponding author upon reasonable request. Source data are provided with this paper.

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

## Acknowledgements
This work was supported by grants from the National Natural Sciences Foundation of China (91642117, 31970840, 31872735, and 31330025).

## Author contributions
Yuqing Wang, W.W., and Y. Zhang designed the project. Yuqing Wang and W.W. did the experiment and Yuqing Wang, W.W., and Y. Zhang wrote the manuscript. X. Qian, H.X., Y.L., and J.H. hybrid the mice. Y.Y., X. Qin and Y. Zhou contributed to establish the tumor models. H.W., J.W., and Z.Z. analyze the data. X.S., Yan Wang, and X.P. contributed to Confocal fluorescence microscopy examination.

## Competing interests
The authors declare no competing interests.

## Ethics statement
This study was carried out in accordance with the recommendations of the Ethics Committee of Peking University Health Science Center. The protocol (No. LA2018106) was approved by the Ethics Committee of Peking University Health Science Center.
