## [Peer Review File · Nature Communications]

Reviewers' comments:

Reviewer #1 (Remarks to the Author):

\In the manuscript entitled 'The essential role of PRAK in tumor metastasis and its therapeutic potential', the authors showed PRAK (also known as MAPKAPK5, MK5) mediated the cancer cell migration into the lung, and consequently lung metastasis. Mechanistically, the authors tried to link PRAK with mTOR1 activation, which led to HIF1a expression; and HIF1a activated MMPs and EMT. The authors validated these findings in clinical samples.

The authors used genetic and pharmacological approaches in 4 mouse models, both xenograft and synergic, of breast cancer and melanoma to show the extremely significant effect of PRAK in lung metastasis. These results made PRAK a potential target to prevent cancer cell seeding in the lung. However, the data to delineate the underlying mechanisms are correlated, but not conclusive. Moreover, the patients' samples/data used for clinical validation were not consistent with experimental models.

Major points:

Point 1: In vitro cell culture experiments: the authors concluded that PRAK contributed to cancer cell migration, but not cell proliferation and apoptosis. However, all these tests were under normal cell culture condition without any stress. Stress conditions need to be applied to retest cell proliferation and apoptosis, especially stress related glutathione metabolism and hypoxia were the top scored pathways in Figure 4B.

For proliferation assays, MTA assay could be biased since this assay depends on cell metabolism while the authors showed mTOR1 activity was regulated by PRAK. For apoptosis assays, apoptotic inducers need to be used to increase cell apoptosis and detect whether different PRAK expression/activity have any effect.

Without proper experimental designs, the authors could not make following claim 'In view of the importance of p38-PRAK signaling in cellular response to stresses, the present study investigated the potential role of PRAK in the adaptation of metastatic tumor cells to a dramatically changed microenvironment.'

Point 2: In vivo lung metastasis animal experiments: the effect of FRAK on overall lung metastasis was extremely impressive. Based on the in vitro data, the only effect of PRAK was migration. Meanwhile, in the tail vein injected lung metastasis model, application of PRAK inhibitor for the first 4 days had same effect as whole course (15 days) treatment while starting treatment from day 5 had absolutely no effect on lung metastases. All these data suggested that PRAK only regulated cancer cell extravasation into the lung. This point needs to be strengthened by the following experiments using the existing tissues.

1) In addition to show one image under each condition, quantifications need to be done in Figure 3D, SA and SB.

2) For the lung metastases (for example, MDA-lung metastases in Figure S3A), the authors should quantify the numbers of lung metastases, and more importantly, the sizes of the lung metastatic lesions. In theory, once the cancer cells get into the lung, FRAK should have no effect on further metastatic outgrowth (the size of lung metastases).

Point 3: The authors showed FRAK regulated cancer cell migration, which should be involved in cancer cell intravasation into the circulation from primary tumors, extravasation into the distal organs, and lymphatic metastasis.

The fact that FRAK did not change primary tumor growth makes it easier to compare CTCs from control and FRAK-depleted primary tumors. If directly detecting CTCs is technically difficult, the authors can indirectly quantify human cancer cells in the whole blood collected from xenograft models (s.c B16 or mammary fat pad implanted MDA primary tumor models) by RT-PCR. The authors can use human specific b2M primers to quantify its mRNA, normalized by b2M expression from mouse blood cells (using mouse specific b2M primers),

Draining LN of the primary tumors could be tested in the animal models, which could make the clinical validation more relevant.

Point 4: The data to delineate the underlying mechanisms are correlated, but non-conclusive. The current data suggested, FRAK activated mTOR1, mTOR1 increased HIF1a, HIF1a increased MMPs and some ECM markers. Every single step needs to be confirmed by at least biochemical approach (western blot) and functional migration assays. Of course, in vivo lung metastasis assay would be more convincing. For example, in FRAP expression/function-deficient cells, add mTOR1 agonist,

detect HIF1a, MMPs, E-, N-cadherin and migration. in FRAP expression/function-deficient cells, overexpress HIF1a, detect MMPs, E-, N-cadherin and migration. The authors did some of suggested experiments. The data were spreading through Figure 4, 5 and S4, S5. It was hard to follow and incomplete.

mTOR1 activation can increase overall translation. Could FRAK activated mTOR1 non-specifically increase HIF1a translation?

Point 5: Clinical validations were comparing primary tumors and LN metastases, instead of lung metastases. In the main figure, the more complete clinical validation was performed in lung cancer patients. Breast cancer samples were in the supplementary figure and there is no melanoma data. It is understandable that clinical materials are hard to obtain. However, please perform similar analysis as Figure 6F in public data of breast cancer and melanoma.

Minor point:

The manuscript was poorly prepared, hard to follow and with many mistakes.

As pointed out earlier, overclaimed 'In view of the importance of p38-PRAK signaling in cellular response to stresses, the present study investigated the potential role of PRAK in the adaptation of metastatic tumor cells to a dramatically changed microenvironment.'

Figure 5B, the 6 samples in each experimental setting should be prepared together and run in the same western gel. It is important to show 10 hour-MG132 treatment works in the NC cells or control condition (increasing HIF1a protein) first before comparing KO cells or FRAK inhibition.

Figure 5A, what was the gel image? From RT-PCR?

Experimental conditions were not consistent in figure legend (Fig 5D) and method. 6h or 8h?

Line 285, should be Figure S6A-7.

Reviewer #2 (Remarks to the Author):

In this manuscript Wang et al suggest that PRAK activity is relevant for the regulation of metastasis. Whereas no effect on tumor growth is described herein, albeit previously described in the literature, Prak deficiency blunts PyMT driven tumors lung metastasis. Similar results are observed when human tumor cell lines are inoculated intravenous. In addition, the authors went on to provide clinical associations that back up their findings. Overall, these are interesting observations that focus on a clear unmet medical need, the treatment of metastasis.

The authors also elaborate on the molecular mechanism that mediate PRAK dependent support of metastasis. The authors show that PRAK loss of function causes inhibition of HIF- α synthesis, most likely via mTOR. Although they fail to explain what activates PRAK signaling downstream of P38 and other pathways, whether this activation is specific of metastatic cells or consistently observe in all tumor cells. Finally, the authors also proposed the use of a potential inhibitor as a mean to modulate PRAK function blunting metastasis in experimental models.

Overall, this is a relevant and comprehensive well-executed collection of data, providing a clear metastasis role for PRAK, across different mouse and human cancer models, of different cancer types. The function of PRAK validated genetically (loss and rescue functions) and chemically and with strong clinical associations. However, there are some sticky points that require some work.

1- Recent data has position p38MAPK signaling as an inhibitor of tumor metastasis as well as its downstream effector, MSK1, mediating metastatic latency. These functions are counterintuitive with that proposed for PRAK in the context of this manuscript. This requires further discussion. In addition, prompts to a critical point: is PRAK activated by default in tumor cells? Is it activated only upon release from the primary tumor? Is its activation dependent on the metastatic stroma? At what level is the control of PRAK exerted if counterbalances the activation of P38.

2- The PRAK inhibitor treatment window of opportunity is not observed at the genetic level, how come? It would strengthen the findings if recapitulated with an inducible knockdown. Is this reflecting a cross-communication with the stroma, a metabolic dependence?

3- In line 279, the authors describe their data as consistent with many other reports. Yet, they reference only one. Please correct.

4- mTOR inhibitors (Everolimus/rapamycin) are clinically approved to treat ER+ advanced metastatic breast cancer. This provides a nice background to test the PRAK inhibitors and get a direct comparison.

5- In addition, it is unclear to what extent PRAK effect is related to the immune system clearance. The authors should at least discuss the concept.

6- The clinical validations are all based on PRAK expression. However, what is relevant is PRAK activity. To this end, it is unclear how the signaling activity is associated beyond the correlation with HIF1a. Further, it is unclear to this reviewer how the Ab used have been validated for IHC. PRAK-depleted and control cell pellets should be used to establish the technique. In addition, digital scoring would be indicated as well as a receiving operating curve used to establish the proper cut off. Finally, is there any association between PRAK activity and outcome in Breast and Melanoma?

Point-by-point Responses to the Reviewers' Comments

Reviewer #1

Point 1: In vitro cell culture experiments: the authors concluded that PRAK contributed to cancer cell migration, but not cell proliferation and apoptosis. However, all these tests were under normal cell culture condition without any stress. Stress conditions need to be applied to retest cell proliferation and apoptosis, especially stress related glutathione metabolism and hypoxia were the top scored pathways in Figure 4B.

For proliferation assays, MTA assay could be biased since this assay depends on cell metabolism while the authors showed mTOR1 activity was regulated by PRAK. For apoptosis assays, apoptotic inducers need to be used to increase cell apoptosis and detect whether different PRAK expression/activity have any effect.

Without proper experimental designs, the authors could not make following claim 'In view of the importance of p38-PRAK signaling in cellular response to stresses, the present study investigated the potential role of PRAK in the adaptation of metastatic tumor cells to a dramatically changed microenvironment.'

Response: These are very good points. To address the reviewer's concerns, we re-examined the effect of PRAK on cell proliferation under normoxic as well as hypoxic (5% O₂) conditions through EdU incorporation. As shown in Figure S2C, although cell proliferation was generally slower under the hypoxic condition, similar percentages of EdU⁺ cells were detected in *Prak*^{+/+} and *Prak*^{-/-} B16 cell cultures or *Prak*^{+/+} B16 cultures with or without addition of the PRAK inhibitor under given conditions (normoxic or hypoxic). In addition, we re-assessed whether the PRAK activity affected apoptosis induced by cisplatin. As expected, treatment with cisplatin resulted in an overall increase in cell apoptosis. The percentage of Annexin V⁺ cells, however, remained at similar levels with or without functional PRAK (Figure S2E). These results suggest that PRAK is not implicated in cell proliferation and apoptosis even under stress conditions.

We agree the original statement is somehow overclaimed. To be more straightforward, the aim of this study was reworded as follows: "In view of the stress imposed on metastatic cells and the importance of p38-PRAK signaling in the stress response, the present study investigated the potential role of PRAK in another important aspect of tumorigenesis – tumor metastasis."

Point 2: In vivo lung metastasis animal experiments: the effect of PRAK on overall lung metastasis was extremely impressive. Based on the in vitro data, the only effect of PRAK was migration. Meanwhile, in the tail vein injected lung metastasis model, application of PRAK inhibitor for the first 4 days had same effect as whole course (15 days) treatment while starting treatment from day 5 had absolutely no effect on lung metastases. All these data suggested that PRAK only regulated cancer cell extravasation into the lung. This point needs to be strengthened by the following experiments using the existing tissues.

1) In addition to show one image under each condition, quantifications need to be done

in Figure 3D, SA and SB.

2) For the lung metastases (for example, MDA-lung metastases in Figure S3A), the authors should quantify the numbers of lung metastases, and more importantly, the sizes of the lung metastatic lesions. In theory, once the cancer cells get into the lung, FRAK should have no effect on further metastatic outgrowth (the size of lung metastases).

Response:

1) In addition to the representative images, quantification on the number of cells or tumor nodules was performed with 5-6 randomly selected fields. The results are shown as additional panels in the corresponding figures.

2) It is a good suggestion. We measured the diameter of 15 randomly selected tumor nodules under each condition, and the results were shown in the lower panel of Figure S3A. Indeed, while *Prak* knockdown and PRAK inhibition led to a dramatic decrease in the number of metastatic nodules, the size of the few nodules generated under these conditions were comparable to the control group or only slightly reduced, supporting that, once metastases were established, PRAK had no or minimal effect on their outgrowth.

Point 3: The authors showed FRAK regulated cancer cell migration, which should be involved in cancer cell intravasation into the circulation from primary tumors, extravasation into the distal organs, and lymphatic metastasis.

The fact that FRAK did not change primary tumor growth makes it easier to compare CTCs from control and FRAK-depleted primary tumors. If directly detecting CTCs is technically difficult, the authors can indirectly quantify human cancer cells in the whole blood collected from xenograft models (s.c B16 or mammary fat pad implanted MDA primary tumor models) by RT-PCR. The authors can use human specific b2M primers to quantify its mRNA, normalized by b2M expression from mouse blood cells (using mouse specific b2M primers),

Draining LN of the primary tumors could be tested in the animal models, which could make the clinical validation more relevant.

Response: Thanks for the suggestion. To test this, 2×10^6 MDA-MB231 cells carrying shSCR or shPRAK were fat pad implanted into SCID mice. Peripheral blood and draining lymph node samples were collected 3 weeks after implantation. RT-PCR was performed using primers specific for human or mouse b2M. The relative ratio of human versus mouse b2M was calculated as $2^{-(\Delta CT(\text{human b2M}) - \Delta CT(\text{mouse b2M}))}$. While we failed to detect human cells in the blood samples, human b2M expression was constantly detected in the draining lymph nodes (see the figure below). However, no significant difference was revealed with the limited number of samples tested so far (5 for shSCR-transfected MDA and 4 for shPRAK-transfected MDA). At this point, we hesitate to formally present this piece of data in the manuscript. It is worth mentioning that, despite the impaired cell migration, two photon microscope examination indicated that PRAK inactivation appear to primarily affect the survival of metastatic cells after migration into distant organs (Figure 3D and Figure S3B).

Point 4: The data to delineate the underlying mechanisms are correlated, but non-conclusive. The current data suggested, FRAK activated mTOR1, mTOR1 increased HIF1a, HIF1a increased MMPs and some ECM markers. Every single step needs to be confirmed by at least biochemical approach (western blot) and functional migration assays. Of course, in vivo lung metastasis assay would be more convincing. For example, in FRAP expression/function-deficient cells, add mTOR1 agonist, detect HIF1a, MMPs, E-, N-cadherin and migration. in FRAP expression/function-deficient cells, overexpress HIF1a, detect MMPs, E-, N-cadherin and migration. The authors did some of suggested experiments. The data were spreading through Figure 4, 5 and S4, S5. It was hard to follow and incomplete.

mTOR1 activation can increase overall translation. Could FRAK activated mTOR1 non-specifically increase HIF1a translation?

Response: Following the reviewer's suggestions, additional experiments were performed and the results are presented as specified below:

The impact of the mTORC1 agonist MHY1485 on migration (Fig. 5G) and HIF-1 α and MMP2/9 expression (Fig. 5F) of PRAK-deficient B16 cells has already been described in the original manuscript. Furthermore, we examined the expression of EMT signature molecules. The results are presented in Fig. S5A. Briefly, MHY1485 effectively reversed the downregulation of N-cadherin and upregulation of E-cadherin induced by *Prak* deletion.

The impact of restored HIF-1 α expression on migration (Fig. 4G) and lung metastasis (Fig. 4H) of PRAK knockdown MDA-MB-231 cells has already been described in the original manuscript. Furthermore, we examined the expression of MMP2/9 and EMT signature molecules, and the results are presented in Fig. S4C and Fig. S4D, respectively. In short, overexpression of HIF-1 α restored the expression of MMP2, N-cadherin and E-cadherin in *Prak* knockdown MDA-MB231 cells.

While mTORC1 activation leads to a global increase in protein translation, a disproportionate and dramatic increase is often observed in the translational rate of a specific subset of transcripts bearing 5'TOP, including HIF-1 α mRNA¹. Therefore, there is some selectivity in PRAK-mediated regulation of HIF-1 translation.

Point 5: Clinical validations were comparing primary tumors and LN metastases, instead of lung metastases. In the main figure, the more complete clinical validation was performed in lung cancer patients. Breast cancer samples were in the supplementary figure and there is no melanoma data. It is understandable that clinical materials are hard to obtain. However, please perform similar analysis as Figure 6F in public data of breast cancer and melanoma.

Response: As suggested by the reviewer, we analyzed the correlation of PRAK expression with overall survival of breast and melanoma patients using data from the Gepia database. A significant correlation was observed for breast cancer (Fig. S7A) but not for melanoma patients (Fig. S7B).

Minor point:

The manuscript was poorly prepared, hard to follow and with many mistakes.

As pointed out earlier, overclaimed ‘In view of the importance of p38-PRAK signaling in cellular response to stresses, the present study investigated the potential role of PRAK in the adaptation of metastatic tumor cells to a dramatically changed microenvironment.’

Response: As described in response to Point 1, this statement was reworded as follows: “In view of the stress imposed on metastatic cells and the importance of p38-PRAK signaling in the stress response, the present study investigated the potential role of PRAK in another important aspect of tumorigenesis – tumor metastasis.”

Figure 5B, the 6 samples in each experimental setting should be prepared together and run in the same western gel. It is important to show 10 hour-MG132 treatment works in the NC cells or control condition (increasing HIF1a protein) first before comparing KO cells or FRAK inhibition.

Response: As required by the reviewer, HIF-1 α expression in *Prak* knockout or inhibitor-treated B16 cells in the presence or absence of MG132 was probed on the same blot. As shown below, levels of HIF-1 α were markedly elevated following MG132 treatment. However, due to the big difference between untreated and MG132-treated cells in HIF-1 α expression, it was difficult to find an appropriate exposure time which allows the detection of HIF-1 α signal in untreated cells without over-exposure in MG132-treated cells. The major point to be made here is that, although the inhibition of protein degradation led to the accumulation of HIF-1 α in all samples, HIF-1 α differential expression was maintained between *Prak*^{+/+} and *Prak*^{-/-} cells and between inhibitor-treated and untreated cells, indicating that mechanism(s) other than HIF-1 α degradation results in the differential expression. As such, we choose to keep the original figures obtained with different exposure times.

Figure 5A, what was the gel image? From RT-PCR?

Response: Figure 5A shows an agarose gel of the RT-PCR products. We further clarified this in the Figure legend.

Experimental conditions were not consistent in figure legend (Fig 5D) and method. 6h or 8h?

Response: Thanks for pointing out the inconsistency. The pulsing was carried out for 6h. The typo has been corrected in the method.

Line 285, should be Figure S6A-7.

Response: The typo has been corrected.

Reviewer #2

- 1、Recent data has position p38MAPK signaling as an inhibitor of tumor metastasis as well as its downstream effector, MSK1, mediating metastatic latency. These functions are counterintuitive with that proposed for PRAK in the context of this manuscript. This requires further discussion. In addition, prompts to a critical point: is PRAK activated by default in tumor cells? Is it activated only upon release from the primary tumor? Is its activation dependent on the metastatic stroma? At what level is the control of PRAK exerted if counterbalances the activation of P38.

Response: Accumulating evidence indicates a complex role of p38MAPK signaling in the control of tumor metastasis. By enforcing metastatic dormancy, it is believed to have an inhibitory effect on tumor metastasis. A recent study by Gawrzak et al., for example, have demonstrated that p38MAPK activates MSK1, which in turn strengthens metastatic latency in ER⁺ breast cancer by promoting luminal cell differentiation². Other studies, however, have reported a pro-metastatic activity of p38MAPK in a wide array of tumors³⁻⁵. These seemingly conflicting data reflect the wide variety of substrates downstream of p38MAPK activation. This point has been further explored in the Discussion.

While PRAK has been shown to be activated by p38MAPK and atypical MAPK ERK3/4 under various stress conditions *in vitro*, little is known about its activation status and the mechanisms governing its activity in tumor tissues, partly due to the unavailability of appropriate antibodies. We are currently working to generate hybridomas secreting antibodies against the phosphorylated PRAK. Once the antibodies are available, we should be able to address many of the important questions raised by the reviewer.

- 2、The PRAK inhibitor treatment window of opportunity is not observed at the genetic level, how come? It would strengthen the findings if recapitulated with an inducible knockdown. Is this reflecting a cross-communication with the stroma, a metabolic dependence? It would strengthen the findings if recapitulated with an inducible

Response: As suggested by the reviewer, we tested the “window of opportunity” at the genetic level using a lentiviral vector expressing shPRAK under the control of a Tet-On promoter. B16 cells bearing the inducible shPRAK construct was intranevously injected into mice. Doxycycline was supplied in the drinking water 1 day before or 4 days after tumor cell injection. Tumor colonies in the lung were counted at day 15. Consistent with previous results obtained with the PRAK inhibitor, inducible knockdown in the first 5 days markedly inhibited lung metastasis, whereas delayed administration of Doxycycline in the last 10 days had virtually no effect (Figure 2D and E).

At this point, we are not sure whether the “window of opportunity” results from a requirement for a cross-talk with the stroma or a cell-autonomous metabolic switch. These are obviously important questions to be addressed in future studies.

- 3、 In line 279, the authors describe their data as consistent with many other reports. Yet, they reference only one. Please correct.

Response: The reference cited here in the previous version was a review paper. In response to the reviewer's comments, three additional original papers have been included in the revised manuscript.

- 4、 mTOR inhibitors (Everolimus/rapamycin) are clinically approved to treat ER+ advance metastatic breast cancer. This provides a nice background to test the PRAK inhibitors and get a direct comparison.

Response: Following the reviewer's suggestion, we directly compared the inhibitory effect of the mTORC1 inhibitor (rapamycin) and the PRAK inhibitor on lung dissemination of the ER⁺ breast cancer cell line MCF7. As shown in Figure S5C, these two inhibitors have similar potency in the metastasis inhibition.

- 5、 In addition, it is unclear to what extent PRAK effect is related to the immune system clearance. The authors should at least discuss the concept.

Response: As similar inhibition was observed in C57BL/6 (for studies of spontaneous breast cancer and B16 melanoma) and SCID (for studies of human melanoma A375 and breast cancer MDB-MB231) mice, we believe that the immune system is unlikely a major contributor to the anti-metastatic effect of PRAK inactivation. This point has been made clear in line 149-152.

- 6、 The clinical validations are all based on PRAK expression. However, what is relevant is PRAK activity. To this end, it is unclear how the signaling activity is associated beyond the correlation with HIF1a. Further, it is unclear to this reviewer how the Ab used have been validated for IHC. PRAK-depleted and control cell pellets should be used to establish the technique. In addition, digital scoring would be indicated as well as a receiving operating curve used to establish the proper cut off. Finally, is there any association between PRAK activity and outcome in Breast and Melanoma?

Response: It is a good point. Phosphorylation of PRAK could be a potential indicator of its activation. However, the anti-phosphorylated PRAK antibody described in previous reports is longer available due to the loss of the hybridoma.

We did validated the suitability of the antibody for IHC by staining parent and *Prak* knockout B 16 cells. The results are shown below.

PRAK and HIF-1 α expression in the tissue arrays was also scored using Image pro plus. Such analyses confirmed the elevated expression of PRAK and HIF-1 α in lymphatic metastases (see figures below).

Lung carcinoma

Breast carcinoma

Moreover, we analyzed the correlation of PRAK expression with overall survival

of breast and melanoma patients using data from the Gepia database. The results are presented in Figure S7. A significant correlation was observed for breast cancer but not for melanoma.

- 1 Mitchell, S. A. *et al.* Identification of a motif that mediates polypyrimidine tract-binding protein-dependent internal ribosome entry.
- 2 Gawrzak, S. *et al.* MSK1 regulates luminal cell differentiation and metastatic dormancy in ER(+) breast cancer.
- 3 Curtis, M. *et al.* Fibroblasts Mobilize Tumor Cell Glycogen to Promote Proliferation and Metastasis.
- 4 Pavan, S. *et al.* HSP27 is required for invasion and metastasis triggered by hepatocyte growth factor.
- 5 Zhang, Y. *et al.* p38-regulated FOXC1 stability is required for colorectal cancer metastasis.

REVIEWERS' COMMENTS

Reviewer #1 (Remarks to the Author):

The authors did the suggested experiments and modification on the text raised by me.

Reviewer #2 (Remarks to the Author):

The authors addressed most of my concerns. Although P_{ARK} activation remains a sticky issue, the manuscript has significantly improve through the review process.